# Deep Learning Supplants Visual Analysis by Experienced Operators for the Diagnosis of Cardiac Amyloidosis by Cine-CMR

**DOI:** 10.3390/diagnostics12010069

**Published:** 2021-12-29

**Authors:** Philippe Germain, Armine Vardazaryan, Nicolas Padoy, Aissam Labani, Catherine Roy, Thomas Hellmut Schindler, Soraya El Ghannudi

**Affiliations:** 1Department of Radiology, Nouvel Hopital Civil, University Hospital, 67000 Strasbourg, France; aissam.labani@chru-strasbourg.fr (A.L.); catherine.roy@chru-strasbourg.fr (C.R.); soraya.elghannudi-abdo@chru-strasbourg.fr (S.E.G.); 2ICube, University of Strasbourg, CNRS, 67000 Strasbourg, France; vardazaryan@unistra.fr (A.V.); npadoy@unistra.fr (N.P.); 3IHU (Institut Hopitalo-Universitaire), 67000 Strasbourg, France; 4Mallinckrodt Institute of Radiology, Division of Nuclear Medicine, Washington University School of Medicine, Saint Louis, MO 63110, USA; thschindler@wustl.edu; 5Department of Nuclear Medicine, Nouvel Hopital Civil, University Hospital, 67000 Strasbourg, France

**Keywords:** cardiac amyloidosis, AL/TTR amyloidosis, hypertrophic cardiomyopathy, left ventricular hypertrophy, deep learning, convolutional neural network

## Abstract

Background: Diagnosing cardiac amyloidosis (CA) from cine-CMR (cardiac magnetic resonance) alone is not reliable. In this study, we tested if a convolutional neural network (CNN) could outperform the visual diagnosis of experienced operators. Method: 119 patients with cardiac amyloidosis and 122 patients with left ventricular hypertrophy (LVH) of other origins were retrospectively selected. Diastolic and systolic cine-CMR images were preprocessed and labeled. A dual-input visual geometry group (VGG ) model was used for binary image classification. All images belonging to the same patient were distributed in the same set. Accuracy and area under the curve (AUC) were calculated per frame and per patient from a 40% held-out test set. Results were compared to a visual analysis assessed by three experienced operators. Results: frame-based comparisons between humans and a CNN provided an accuracy of 0.605 vs. 0.746 (*p* < 0.0008) and an AUC of 0.630 vs. 0.824 (*p* < 0.0001). Patient-based comparisons provided an accuracy of 0.660 vs. 0.825 (*p* < 0.008) and an AUC of 0.727 vs. 0.895 (*p* < 0.002). Conclusion: based on cine-CMR images alone, a CNN is able to discriminate cardiac amyloidosis from LVH of other origins better than experienced human operators (15 to 20 points more in absolute value for accuracy and AUC), demonstrating a unique capability to identify what the eyes cannot see through classical radiological analysis.

## 1. Introduction

Cardiac amyloidosis (CA) is a specific cardiomyopathy caused by the deposition of misfolded amyloid fibrils in the extracellular myocardial space. Light-chain (AL) and transthyretin (TTR) are the most common subtypes. Cardiac amyloidosis is a fatal disease requiring rapid diagnosis for patients to benefit from recently released medications [1,2,3]. Its diagnosis has gained significant improvements in recent years, in particular with the recognition of diphosphonate SPECT imaging for the identification of the TTR form of the disease [4].

MRI plays an important role in this field thanks to gadolinium injections providing quite a specific pattern of myocardial late-enhancement [5] and demonstrating highly relevant extracellular volume (ECV) increase [6]. Despite recent relief in the restrictions on the use of gadolinium chelates [7], caution needs to be exercised in case of renal impairment, and a diagnostic approach without injection would be beneficial. Steady-state free precession (SSFP) cine-CMR is a basic method in cardiac MRI, offering a good quality morphological and functional depiction of important cardiac features [8]. Myocardial wall thickening, atrial enlargement and pericardial or pleural effusion constitute the hallmarks of amyloid cardiac involvement [9]. However, these signs are very nonspecific since they are also seen in many other etiologies of left ventricular hypertrophy such as advanced hypertensive disease, aortic stenosis and other overload diseases such as Fabry disease and sarcomeric hypertrophic cardiomyopathies, which is why cine-CMR alone is not recognized as effective for diagnosing cardiac amyloidosis.

Machine learning and, particularly, deep learning applied to imaging quickly established themselves in most pathological areas, and these methods are now recognized as having diagnostic capacities similar to experienced radiologists, particularly in cardiomyopathies [10] and cardiac amyloidosis [11]. An even more interesting fact concerns the superior diagnostic capacities of deep learning over human readers in some fields, such as breast cancer [12], especially its ability to identify pathologies invisible to the naked eye, such as abnormalities discernible only in immunohistochemistry or through genetic analysis. For example, deep learning was reported to be efficient in improving mutation prediction in hypertrophic cardiomyopathy using MR-cine images [13].

This innovative concept led us to initiate the present study in which we compared the performance of commonly available deep learning methods to experienced radiologists to discriminate cardiac amyloidosis from other myocardial hypertrophies based on cine-CMR alone. Moreover, we explored the capacity of deep learning to differentiate AL from TTR amyloidosis, which is not reliably achievable visually with cine-CMR.

## 2. Materials and Methods

### 2.1. Study Population

We retrospectively analyzed the cine-CMR sequences of patients performed between 2010 and 2020 in the radiology department of our hospital. This study was registered and approved by the Institutional Review Board of our university hospital, and all datasets were obtained and de-identified, with waived consent in compliance with the rules of our institution. The cine-CMR exams of 241 patients were studied, including 119 with histologically proven amyloidosis and 122 with left ventricular hypertrophy without amyloidosis (LVH). The patients’ characteristics are listed in Table 1.

The left ventricular hypertrophy without amyloidosis group (*n* = 122) consisted of patients referred to CMR for suspected cardiac amyloidosis due to several suggestive features such as a heart failure episode, thickening of the myocardial walls on ultrasound examination, restrictive transmitral Doppler filling pattern, reduced longitudinal strain with apical sparring, monoclonal gammopathy or dubious Perugini grade 1 bone scintigraphy. Other cases presented a CMR of concentric left ventricular hypertrophy (left ventricular wall thickness ≥13 mm in diastole). The clinical context was consistent with hypertension, aortic stenosis or non-obstructive hypertrophic cardiomyopathy. Late-enhancement imaging obtained in all cases never demonstrated circumferential subendocardial or diffuse late-enhancement patterns suggestive of amyloid involvement.

For the amyloidosis group (*n* = 119), the selection criteria for amyloidosis diagnosis were based on typical CMR features confirmed by clinical, biological, bone scintigraphic and anatomo-histological findings. Left ventricular wall thickening (≥13 mm in diastole), left ± right atrial dilatation, increased native myocardial T1 relaxation time and/or extracellular volume (ECV), pericardial or pleural effusion and typical subendocardial late-enhancement pattern (circumferential, diffuse or not related to a coronary territory) were the main diagnostic clues for amyloidosis. 

The characteristics of AL and TTR patients can be found in the supplemental material (Appendix A). TTR amyloidosis was defined in 38 patients without monoclonal gammopathy and with a ^99m^Tc-diphosphonate SPECT Perugini score of >1 or with amyloid deposits on an extracardiac and/or endomyocardial biopsy. AL amyloidosis was reported in 59 cases, based on the detection of a kappa/lambda free light-chain with monoclonal gammopathy and an extracardiac and/or endomyocardial biopsy. Among the 22 patients who were not categorized as AL or TTR, three were AA type, three had uncertain immunostaining, one had Perugini 1 and no gammopathy, four elderly patients died and 11 were lost to follow-up.

For the cine-CMR acquisitions, all images were obtained at 1.5 Tesla, using three Siemens (Erlangen, Germany) and one Philips (Eindhoven, The Netherlands) scanners. Steady-state free precession (SSFP) cine sequences were obtained with TE/TR 1.6/3.5 ms, 8 to 32 elements cardiac coil and 6 to 8 mm thick slices. End-systole (with the smallest left ventricular dimension) was visually selected (systolic time in Table 1). Orientation planes were long axis (4-chamber and vertical 2-chamber views) and short axis views. Table 1 lists the summary of acquisition parameters.

### 2.2. Image Preparation

The image preparation of cine studies exported from the PACS of our hospital was carried out with a dedicated Visual C software. All images were first de-identified and resampled (bilinear) in order to obtain a normalized homogeneous pixel size of 1.5 mm. The images’ intensity windowing was manually focused on the central cardiac region of interest. Diastolic and systolic frames were selected. Epicardial contours (ROI_epi) and myocardial contours (ROI_myo) were manually drawn. 

Finally, five pairs of images (cropped to 128 and 160 pixels, full view 256 pixels, ROI_epi and ROI_endo), as illustrated in Figure 1, were stored. The purpose of these tests (especially for ROIs) was to determine if a focused analysis led to better classification performance. Labeling (orientation plane, pathology, presence of effusion and gadolinium injection) was carried out simultaneously and saved in the labeled file.

### 2.3. Deep Learning Process

CNN implementation was performed in Python 3.7.6, with Keras library and TensorFlow backend. According to CLAIM recommendations [14], the data were distributed in order to ensure that images of the same patient always lie in either the train set, the validation set or the test set (no mixture between these sets). 

For hyperparameter trimming, data processing was performed according to the diagram shown in Figure 2. A 40% test set (538 pairs of frames and 96 patients) was isolated and stored as a held-out test set. With the 60% remaining data, a three-fold cross-validation training was performed in order to trim hyperparameters (batch size, optimizer, learning rate, decay, number of trainable layers, dropout rate and parameters of the image data generator). This was done to avoid the influence of individual training and validation examples on the choice of hyperparameters. 

With optimal hyperparameters, a final model was built with all training data and evaluated on the test set. Patient-based metrics were calculated from the average of the predicted probability corresponding to all frames of a unique patient.

A VGG16 [15] base model was used and trained from scratch for diastolic and systolic frames. The two outputs (diastole and systole) were concatenated and followed by the following layers, where ReLU non-linearity was used after each Dense layer except the last one: Flatten, Dense 256, Dropout 0.40, Dense 128, Dropout 0.45, Dense 64, Dropout 0.50, Dense 1 and output Sigmoid activation layer. In the final model, training was done with batch size: 32; number of epochs: 150; optimizer: SGD; LR 4 10^−5.^; and decay: 10^−6^. Binary cross entropy was used as a loss function. The parameters of data augmentation applied during training were zoom range <0.15, 15% height and width shift range and up to 20° rotation.

The Grad-CAM algorithm [16] was used to visualize class activation maps. With this algorithm, the identification of the most contributive pixels involved for each class is related to the gradient information flowing into the final convolutional layer of the network.

### 2.4. Experienced Radiologists/Cardiologists Blind Reading 

The blind reading of diastolic and systolic images was performed by one radiologist and two cardiologists (>10 years’ experience of CMR analysis and reporting). Frame-based reading was obtained from the pairs of images corresponding to the test set. Patient-based reading was obtained from the whole dataset (241 patients), and paired comparisons were made with the 40% held-out test set (average number of frame pairs, 5.5 per patient). 

### 2.5. Evaluation and Statistical Analysis

The performance metrics—computed on a frame-basis and a patient-basis—were test accuracy, sensitivity, specificity, confusion matrices, receiver operating characteristic (ROC) curves and precision-recall curves with the corresponding area under the curve (AUC) values. Testing the relationship between categorical variables (e.g., accuracy comparisons) was carried out with a Chi-square test. A comparison of the quantitative values was performed with Student’s *t*-test, and a comparison of the AUC of ROC curves was performed with the Delong test. MedCalc 12.1.4 (MedCalc Software, Ostend, Belgium) was used for statistical analyses.

## 3. Results

### 3.1. Amyloidosis vs. LVH Classification Obtained with the Held-Out Test Set According to the Input Shape 

Table 2 lists the results obtained with the various input shapes illustrated in Figure 1. Patient-based results were always better than frame-based results. 

Optimal performance was obtained with 160 × 160 cropped diastolic and systolic images in which per frame analysis provided a test accuracy of 0.759 and an AUC of 0.836, whereas per patient analysis provided a test accuracy of 0.812 and an AUC of 0.937. 

Combining diastole and systole did not improve the results. Full field 256 × 256 frames and focused myocardial ROI images provided significantly weaker results.

### 3.2. Amyloidosis vs. LVH Classification Obtained with the Held-Out Test Set by Human Readers and by CNN 

The comparison between classification by experienced radiologists/cardiologists and the CNN is given in Table 3. The CNN provided a largely superior performance when compared to human readers. 

Frame-based comparisons of human vs. CNN classification led to an accuracy of 0.605 vs. 0.746 (*p* < 0.0008) and an AUC of 0.630 vs. 0.824 (*p* < 0.0001). 

Patient-based comparisons provided an accuracy of 0.660 vs. 0.825 (0.008) and an AUC of 0.727 vs. 0.895 (*p* < 0.002). The ROC curves of these comparisons are plotted in Figure 3.

### 3.3. CNN Classification of AL vs. TTR Amyloidosis 

The frame-based accuracy and AUC obtained by the CNN classification of AL vs. TTR cardiac amyloidosis were 0.662 and 0.703 [0.664–0.741]. The corresponding patient-based values were 0.711 and 0.752 [0.654–0.834]. No comparison was performed here with human classification, but the comparison between the AUC values of the CNN and the simple left ventricular septal wall thickness measurement (per-patient AUC 0.735) did not show a statistically significant difference. 

### 3.4. Analysis of the Saliency Maps 

Saliency maps, which reveal the pixel areas responsible for classification, show that cardiac regions contribute to CNN decisions in only 25% of cases (Figure 4). Among the extracardiac targeted regions, the lungs are the most frequent, followed by the subcutaneous fat and liver. Distribution is quite similar for correct classification (concordant) and erroneous classification (discordant).

## 4. Discussion

The most important result of this study is the possibility of discriminating cardiac amyloidosis and LVH from other origins by simple cine-CMR images, which was significantly better with the CNN than by the physicians’ visual analysis. The comparison carried out on slightly more than 100 patients of the two groups shows that, for frame-based and patient-based analysis, binary classification accuracy is approximately 15 absolute points higher with the CNN than with experienced radiologists/cardiologists. The same significant difference is also found by considering the AUC of the ROC curve, with a little less than 20 points absolute value improvement with the CNN as compared with experienced human readers. 

### 4.1. Methodological Considerations

Patient-based analysis constitutes a much more relevant assessment because this is how the clinical diagnosis is carried out, and it should be noted that the transposition of the results from the image-level to the patient-level (by taking the average of the elementary predictions per frame) leads to an improvement of the accuracy in the range of 5 points (absolute value) and in the range of 10 points for the AUC. This phenomenon, which is observed for the human reader and CNN, may be explained thanks to the “averaging process” in the mind of the physician who examines the whole set of pictures of the patient. 

The influence of methodological choices should be stressed: (1) The distribution of patients’ images in a distinct train or validation/test data sets is mandatory; otherwise, the results would be clearly biased because we would have trained on images that are—for some features—similar to test images. Processing this way (without frame distribution for a unique patient) with our data set led to a misleading “improvement” of almost 10 points (absolute value) for accuracy and AUC results (data not listed here). (2) The strict separation of the train and validation set for hyperparameter tuning and the test set has been done. This method, based on a separate test set, schematized in Figure 2, is required to avoid information leakage related to hyperparameter tuning. 

### 4.2. Superiority of CNN Capacities over Human Diagnosis 

The aim of this study was not to propose making the diagnosis of cardiac amyloidosis solely on the cine-CMR data because much more relevant CMR indices are available thanks to gadolinium injection. Actually, late-enhancement and ECV allow the diagnosis of the presence of CA with a high sensitivity of 95% and an even higher specificity of 98% [5], and deep learning was demonstrated to be efficient in this field [11]. Our goal was to show that deep learning is able to extract diagnostic clues clearly surpassing visual analysis (15 to 20 points in the present study).

Excellent performances of the CNN are often reported in the literature, but their interest is limited if they are not compared to human performance. Among human–machine comparisons, many studies have reported that CNN diagnosis is on par with human visual assessment in multiple areas [17]. For example, for malignancy risk estimation of pulmonary nodules using thoracic CT, Venkadesh et al. [18] reported that the DL algorithm had an AUC of 0.96, which was significantly better than the average AUC of the clinicians (0.90) but comparable to that of thoracic radiologists. Our model was able to discriminate between AL and TTR CA with interesting values of patient-based accuracy (0.711) and AUC (0.752); however, this was no better than the classification obtained with the simple measurement of the septal thickness, already reported in previous publications [19,20,21] and resulting from the known increased amyloid burden in this subtype.

Of more interest is to show significant machine-over-human superiority in routine areas, where “clinical” visual analysis is the classic benchmark. Our study provides an interesting demonstration in this direction for diagnosing cardiac amyloidosis from cine-CMR. A small number of other publications could demonstrate that AI systems are capable of surpassing human experts in disease prediction. Such is the case for the distinction between low-grade and high-grade glioma by radiologists, which lacks accuracy (40–45% of non-enhancing MR lesions are found subsequently to be malignant glioma), whereas, in contrast, CNN-based grading provides > 90% accuracy [22]. Resnet-50 CNN outperformed 136 of 157 dermatologists in a head-to-head dermoscopic melanoma image classification task [23]. For the diagnosis of breast cancer, in a large multicenter study, Mc Kinney et al. [12] found that the AI system exhibited specificity and sensitivity superior to that of radiologists practicing in an academic medical center and exceeded the average performance of radiologists by a significant improvement in the area under the ROC curve (ΔAUC = +0.115). Similarly, in differentiating benign from malignant renal tumors, Xu et al. [24] reported higher AUC with the CNN model (0.906, based on T2-weighted images) as compared to the AUC obtained by two radiologists (0.724). 

### 4.3. Unveiling the Invisible

One more step in this diagnostic quest is the possibility of discriminating pathological conditions that clinicians are not able to predict at all using the naked eye. Subtyping molecular markers, histological or immune-histochemical and genetic classes is impossible to ascertain from radiologic data. These identifications were initially proposed from radiomic signatures, for instance, to discriminate between hypertensive heart disease and hypertrophic cardiomyopathy [25] or between recent infarction vs. old infarction [26]. However, several comparative studies have demonstrated that deep learning based on radiologic data is superior to radiomics. This has been demonstrated for renal cancer [24], subtyping different types of cerebral glioma [27], diagnosis of breast cancer [28] and predicting axillary lymph node metastasis of breast cancer [29]. 

This may be explained because radiomics’ features are handcrafted in advance and, thus, may not always fit to discriminate particular tasks. In contrast, the CNN is more flexible, adaptive and dynamic. As a data-driven tool, it is able to automatically learn to extract and select task-specific features if the amount of training data is large enough. Further evidence for the power of deep learning to make a histological diagnosis from radiological data has been provided by Zhao et al. for renal cell carcinomas Fuhrman-grading [30] and by Yuan et al. for prostate cancer Gleason score staging (accuracy 0.87) [31].

### 4.4. Explanation of Classification Remains Unsatisfactory

Deep neural networks operate through a multilayer nonlinear structure, making their predictions difficult to interpret. They are able to pick up a number of features that cannot be interpreted by humans but which are relevant for making a diagnosis. These automatically-learned discriminative features are unfortunately presently not clearly identifiable. 

Grad-CAM helps identify the areas of pixels that are most responsible for class prediction [16]. This should provide valuable clues to understand the algorithm’s decision. In principle, the salient areas should be located in the cardiac region, which only appeared in a quarter of the cases in our study. Two explanations may be advanced for this anomaly.

(1) Technically, our network uses only fully connected layers in the last phase, which is where the classification happens, but saliency cannot be obtained from fully connected layers. As a solution, we should try replacing some of the fully connected layers that come right after VGG with convolution. This way, the spatial information would be preserved longer in the network, and we might see more meaning in the saliency maps. 

(2) Amyloidosis is not a disease confined to the heart since the involvement of the lungs, fatty tissues and other organs is also common. Liver and, moreover, spleen amyloid deposits have been reported in 41% of patients with systemic amyloidosis (almost only in AL type), and CMR-derived ECV measurement showed good diagnostic capability in this field [32]. This is why the diagnosis is also based on extracardiac biopsies, and it is interesting to note that the texture analysis was able to show specificities in the architecture of ultrasound images within abdominal fat [33], resulting in increased echogenicity and a loss of the normal structure of the fat layer, consistent with histopathological amyloid deposition in the fat.

This ubiquitous aspect of the disease may also explain why the input shape submitted to the CNN (from the full field image to the small region of interest focused on the sole myocardium illustrated in Figure 1) hardly modifies the performance of our model as shown in Table 2. It can also be noted in Table 2 that the combination of diastole and systole does not provide any diagnostic benefit, unlike for other cardiomyopathies [10], because the global LV systolic function is generally preserved in the early stage of amyloidosis. 

### 4.5. Study Limitations 

Two types of confounding factors must be mentioned. First, plane orientation and the presence of gadolinium in the sets of images could have influenced the results, but Table 1 shows a perfect equivalence between the two groups. Second, the presence of pericardial or pleural effusion constitutes a more important bias because the prevalence (slightly higher than in the study of Binder et al. [9]) is very different in the two groups. Pericardial effusion is observed in almost 50% of CA, i.e., two times more often than in hypertrophies unrelated to amyloidosis. Pleural effusions are observed in just over a third of CA, i.e., four times more than in other hypertrophies, and mixed effusions are 10 times more frequent in the amyloidosis group than in the LVH group. This disparity probably contributes to the classification made by CNNs (although heat maps rarely focus on areas of effusion) but also influences clinical judgment, so that the bias is the same for the machine and for the human, which, therefore, does not explain the diagnostic superiority of the algorithm.

A multiparametric approach is needed. Only cine-CMR data has been used here, and it is likely that one could significantly improve performances by combining the analysis with other CMR sequences such as T1 mapping, ECV assessment and late gadolinium-enhancement imaging. Based on gadolinium-enhanced images—and not on cine-MR images—Martini et al. obtained an accuracy of 0.88 and AUC of 0.98 [11], but remember that our aim was not to develop the best model to optimize cardiac amyloidosis diagnosis but to compare CNN and human reader performance. For the distinction between AL and TTR CA, it has been reported that transmural patterns of late gadolinium enhancement may differentiate these two types of the disease [21] but with relatively low performance. Recently, the use of a logistic regression model integrating T2 mapping (slightly increased in the AL subtype) and right ventricular ejection fraction combined with age was reported to discriminate between these two subtypes with an AUC of 0.92 [34]. The performance of AI integrating such multiparametric CMR features, especially for the distinction between AL and TTR cardiac amyloidosis, should be explored in the future.

Technical improvements should be implemented. The leverage of more sophisticated CNN models (not limited to the classical VGG model used here) and, moreover, the combination (concatenation) of several multiparametric inputs, with possible additional categorical clinical input variables (e.g., [30]), should improve performance. Orientation plane specific models [11] should also be tested since images of different views were classified here by the same network, which makes learning relevant features from images potentially much harder as it increases variability unrelated to any disease. Significant work also remains to be done to improve the explainability of the results. Finally, the relatively limited number of observations and the monocentric nature of this study constitute another limitation. Multicenter studies could be of interest for the further validation and generalization of our findings.

## 5. Conclusions

In this study, based on cine-CMR images alone, we could demonstrate the ability of CNNs to discriminate cardiac amyloidosis from LVH of other origins significantly better than experienced human operators. The diagnostic accuracy and AUC were 15 to 20 points higher (in absolute value) for the VGG convolutional network used here than for human readers. This diagnostic superiority of the CNN results from the unique capability of the algorithm to identify features invisible to the naked eye, indiscernible through the classical radiological analysis. This scientific novelty, already reported in a few recent articles concerning other pathological fields, opens up promising prospects for improving diagnostic capacities in routine clinical practice. The astonishing potential of CNNs to improve the recognition of pathologies that are imperfectly detectable in radiology and to reveal invisible clues such as the histological type of lesions will certainly constitute a large field of future research.

## Figures and Tables

**Figure 1 diagnostics-12-00069-f001:**
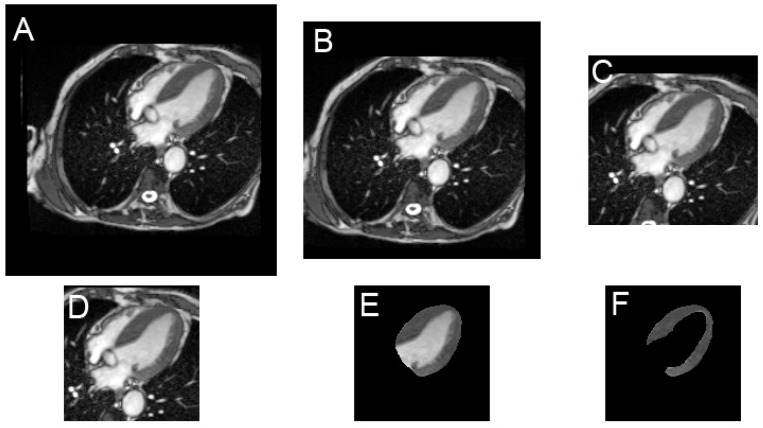
Example of input shapes submitted to the CNN, with native 256 × 256 full image format (**A**), 224 × 224 cropped image (**B**), 160 × 160 cropped image (**C**), 128 × 128 cropped image (**D**), epicardial region of interest (ROI) image (**E**) and myocardial ROI (**F**).

**Figure 2 diagnostics-12-00069-f002:**
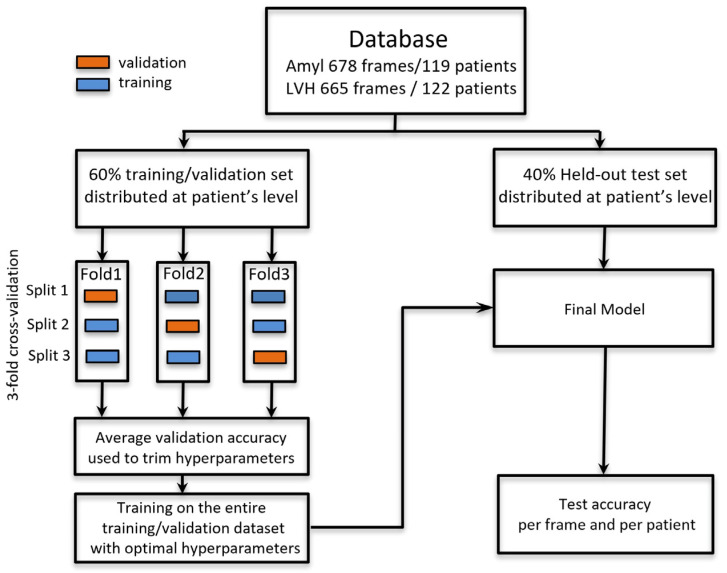
Schematic view of the processing method used in order to strictly separate training/validation data and test data.

**Figure 3 diagnostics-12-00069-f003:**
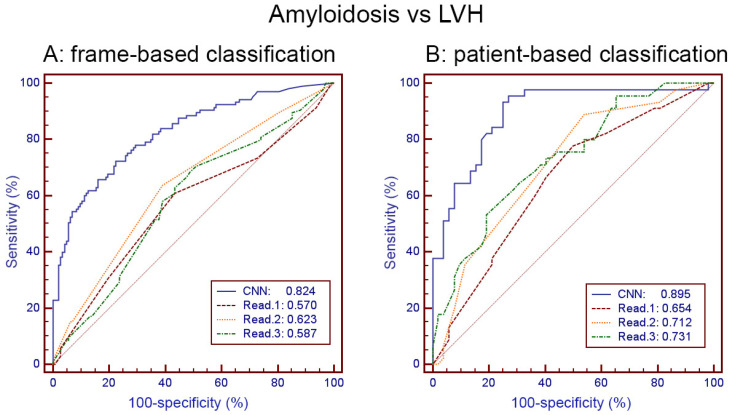
ROC curves and AUC for frame-based (**A**) and patient-based (**B**) classification of amyloidosis vs. LVH by CNN and by three human readers (Read. 1 to 3).

**Figure 4 diagnostics-12-00069-f004:**
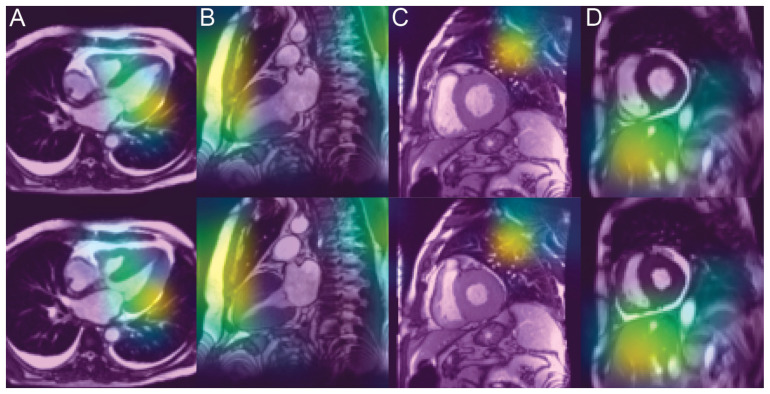
Saliency maps targeting cardiac region (**A**) but also frequently subcutaneous fat (**B**), lung (**C**) or liver (**D**). Diastolic frames are shown in the upper row and systolic frames in the lower row.

**Table 1 diagnostics-12-00069-t001:** Clinical and CMR characteristics of the study population.

	Amyloidosis	LVH	*p*
N patients	119	122	
Age (years)	74.65 ± 9.53	59.50 ± 14.34	0.0001
Sex (F/M)	31/88	39/83	0.31
Weight (kg)	70.80 ± 15.16	82.95 ± 20.50	0.0001
Height (m)	169.9 ± 8.84	170.36 ± 10.05	0.78
BSA (m^2^)	1.84 ± 0.22	2.00 ± 0.27	<0.0001
IVS (mm)	18.11 ± 3.54	18.38 ± 3.54	0.56
LVMI (g/m^2^)	115.96 ± 29.08	116.58 ± 31.43	0.88
LVDVI (mL/m^2^)	69.88 ± 22.21	74.51 ± 20.82	0.36
LVEF (%)	58.96 ± 10.93	67.33 ± 12.18	<0.0001
LA surface (cm^2^)	31.55 ± 5.23	25.47 ± 5.96	0.0002
Systolic time (ms)	321 ± 39	332 ± 40	0.095
T1 (ms)	1138.5 ± 48.1	1038.0 ± 56.2	<0.0001
ECV (%)	53.97 ± 11.17	26.89 ± 4.00	<0.0001
N long axis frames/patient	2.24 ± 0.93	2.22 ± 0.94	0.93
N short axis frames/patient	3.41 ± 1.45	3.59 ± 1.27	0.49
N frames/patient	5.68 ± 1.85	5.47 ± 1.81	0.58
N frame post-gadolinium	171/676	167/667	0.96
N patient with pericard.	54 (45%)	27 (22%)	0.00013
N patients with pleural.	45 (38%)	10 (8%)	0.00001
N patients with both.	24 (20%)	3 (2.5%)	0.00001

The characteristics of patients with amyloidosis and left ventricular hypertrophy were included in this study. The number of observations, (integer) or average values ± standard deviation, are listed: BSA: body surface area; IVS: interventricular septum thickness; LVMI: left ventricular mass index; LVDVI: left ventricular diastolic volume index; LVEF: left ventricular ejection fraction; LA: left atrial; systolic time: the time of the systolic image; and ECV: extracellular volume. Between the parentheses is the percentage. Pericard. is for pericardial effusion, pleural is for pleural effusion and both is for pericardial + pleural effusions.

**Table 2 diagnostics-12-00069-t002:** Accuracy and AUC of the ROC curve for classification of amyloidosis vs. LVH in the 40% held-out test group, according to the input shape.

	Frame-Based		Patient-Based	
Input Shape	Accuracy	ROC AUC	Accuracy	ROC AUC
160 × 160/D + S	0.759	0.836 [0.786–0.878]	0.812	0.937 [0.828–0.987]
160 × 160/D	0.760 (ns)	0.820 (ns) [0.769–0.864]	0.833 (ns)	0.918 (ns) [0.802–0.978]
160 × 160/S	0.733 (ns)	0.801 (0.04) [0.749–0.848]	0.833 (ns)	0.890 (ns) [0.767–0.962]
256 × 256/D + S	0.710 (ns)	0.790 (0.03) [0.735–0.836]	0.771 (ns)	0.803 (0.02) [0.663–0.904]
224 × 224/D + S	0.728 (ns)	0.823 (ns) [0.772–0.867]	0.812 (ns)	0.852 (ns) [0.720–0.938]
128 × 128/D + S	0.740 (ns)	0.808 (ns) [0.756–0.853]	0.812 (ns)	0.922 (ns) [0.807–0.979]
Epicardial ROI	0.722 (ns)	0.787 (0.01) [0.762–0.810]	0.791 (ns)	0.888 (ns) [0.839–0.927]
Myocard. ROI	0.662 (0.05)	0.719 (0.01) [0.693–0.745]	0.714 (ns)	0.814 (0.03) [0.756–0.863]

Results obtained with the 40% held-out test set after hyperparameters tuning. 160 × 160 indicates the cropping size of input frames. D and S indicate diastole and systole. Between brackets is the confidence interval of AUC. Values between parentheses indicate the level of significance of the difference as compared to the 160 × 160 D + S result (assessed with Chi-square test from the number of observations for accuracy and assessed by Delong test for AUC comparisons).

**Table 3 diagnostics-12-00069-t003:** Accuracy and AUC of the ROC curve for classification of amyloidosis vs. LVH in the held-out test group for human readers vs. CNN.

	Frame-Based			Patient-Based		
Metric	Accur.	Sensitiv. Specific.	ROC AUC	Accur.	Sensitiv. Specific.	ROC AUC
CNN	0.746	77.0 71.0	0.824 [0.770–0.869]	0.825	85.7 77.6	0.895 [0.816–0.948]
Read 1	0.585 (0.001)	66.4 50.85	0.570 [0.506–0.632] (0.0001)	0.629 (0.004)	67.4 58.8	0.654 [0.550–0.747] (0.001)
Read 2	0.623 (0.005)	69.6 54.5	0.623 [0.560–0.684] (0.0001)	0.649 (0.009)	69.6 60.8	0.712 [0.611–0.799] (0.0002)
Read 3	0.585 (0.001)	66.4 50.9	0.587 [0.523–0.649] (0.0001)	0.660 (0.013)	71.1 61.5	0.731 [0.631–0.816] (0.002)
Read (avg)	0.605 (0.0008)	69.2 52.7	0.630 [0.567–0.690] (0.0001)	0.660 (0.008)	72.1 61.1	0.727 [0.627–0.813] (0.002)

Frame-based and patient-based results obtained with the held-out test set by human readers and by CNN. Accur. is for accuracy, Sensitiv. and Specific. are for sensitivity and specificity. Values between parentheses indicate the level of significance of the difference between human reader and CNN (assessed with Chi-square test from the number of observations for accuracy and assessed by Delong test for AUC comparisons).

## Data Availability

The database and code can be made available by reasonable request after the agreement of the Clinical Research Department of our hospital.

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
