# Peer review of "Deep Learning Supplants Visual Analysis by Experienced Operators for the Diagnosis of Cardiac Amyloidosis by Cine-CMR"

_diagnostics, 2021, doi:10.3390/diagnostics12010069_

Round 1

Reviewer 1 Report

The authors presented the paper "Deep learning supplants visual analysis by experienced operators for the diagnosis of cardiac amyloidosis by cine-CMR"

1) I haven't seen any significant discussion about the information in Tables 1 and 2. Is it so necessary for the main part of the paper? Maybe, better to present such data in SI?

2) You have a good comparison between the experienced radiologist and CNN results. However, it will be excellent to present the comparison of the previously obtained results by other research groups if it is possible.

3) Conclusion section can be improved by presenting the advantages of the work, a further research recommendation, the clinical significance of your method. Also, It will be good to highlight the scientific novelty.

Minor comments

1) VGG and AUC abbreviations are not decrypted in the abstract.

2) Fig 4. What are the differences between the first and second raw of the pictures? Can you mention it in the picture description?

Author Response

December 24th 2021

We warmly thank the referees for their work and for their helpful comments. We have carefully considered the advices and remarks of the reviewers. Please find below (in blue color) our point-by-point answer to each comment.

Referee 1

Comments and Suggestions for Authors

The authors presented the paper "Deep learning supplants visual analysis by experienced operators for the diagnosis of cardiac amyloidosis by cine-CMR"

1) I haven't seen any significant discussion about the information in Tables 1 and 2. Is it so necessary for the main part of the paper? Maybe, better to present such data in SI?

Table 1 lists mains characteristics of patients in the amyloidosis group and in the hypertrophic group without amyloidosis. This table is discussed in the ‘4.5 Study limitations’ section, pointing the fact that several variables could have an influence on the results. In particular, orientation planes, the number of frames and the percent of cine images obtained after gadolinium injection are not significantly different between the two groups studied. In contrast, pericardial or pleural effusions were largely more frequently observed in the amyloidosis group. This discrepancy is analyzed in the discussion (lines 379 to 391). Thus, we suggest to keep Table 1 in the article.

Table 2 lists the main characteristics of patients with AL and TTR cardiac amyloidosis (CA). Since the distinction between these two forms of amyloidosis was not the primary focus of the study and since these data were not discussed, table 2 has been removed from the article in accordance with suggestion of the referee and presented as supplemental material.

2) You have a good comparison between the experienced radiologist and CNN results. However, it will be excellent to present the comparison of the previously obtained results by other research groups if it is possible.

We agree that the comparison of the performance obtained by CNN and by experienced radiologists for other diseases is an important point of the discussion. Data from the literature concerning this question are reported in two sections of the discussion:

The section “4.2 Superiority of CNN capacities over human diagnosis” reports 5 of these studies, where visual diagnosis is possible but less well obtained than with CNNs

- pulmonary nodules, Venkadesh ref 18 à humans AUC 0.96 vs CNNs 0.90.

- gliomas, ref 22 à humans lack 40-50% malignant glioma vs CNN accuracy > 90%.

- melanoma, ref 23 à Resnet-50 CNN outperformed 136 of 157 dermatologists.

- breast cancer, Mc Kinney ref 12 : ΔAUC = +0.115 in favor of CNNs.

- malignancy of renal tumors, Xu ref 24 : à humans AUC 0.72 vs CNN AUC 0.90

The following section “4.3 Unveiling the invisible” deals with an even more interesting potentiality of CNNs, able to identify pathological features that are absolutely impossible to discern from radiological visual analysis (8 references).

Further comparisons could probably be found in the literature, but they do not appear to be numerous.

3) Conclusion section can be improved by presenting the advantages of the work, a further research recommendation, the clinical significance of your method. Also, It will be good to highlight the scientific novelty.

The conclusion has been modified, emphasizing the clinical significance and advantages of the work, scientific novelty and further research projects,

Minor comments

1) VGG and AUC abbreviations are not decrypted in the abstract.

VGG and AUC acronyms have been explained in the text (abstract).

2) Fig 4. What are the differences between the first and second raw of the pictures? Can you mention it in the picture description?

Thank you for pointing out the missing explanation in this figure. The 1st and second rows in figure 4 have been explained in the caption (diastole and systole).

Referee2

Comments and Suggestions for Authors

The manuscript entitled ‘Deep learning supplants visual analysis by experienced operators for the diagnosis of cardiac amyloidosis by cine-CMR’ utilizes a convolutional neural network (CNN) to discriminate cardiac amyloidosis from left ventricular hypertrophy (LVH) of other origin. Additionally, the authors have compared their findings with visual diagnosis assessed by experienced human operators. Overall, the manuscript is sound at conceptualization and writing part and the findings of the study are interesting. However, some minor concerns should be addressed by the authors before any possible consideration of this manuscript to be published in the journal ‘diagnostics.

  • In #Table 1 and table 2, if authors have taken the average value of some of the characteristics of sample population (for example, Weight and Height). It should be mentioned as ‘average weight’ and ‘average height’ in the tables.

All the values written in the form m ± sd correspond to the mean ± standard deviation, which has been specified in the legend of the table. Integer values correspond to the number of observations.

  • Why the size of the images are not same in #Figure 1? The ' Image preparation ' section should mention the reason for cropping the image and taking the pictures of different magnifications.
  • In fact, it is not the size of the images but their cropping (shape) that has been modified. The purpose of these tests (especially for ROIs) was to determine if a focused analysis led to better performance (this explanation has been added in the “image preparation section”). The penultimate sentence of section 4.4 explains the interest of these tests with regard to the unfocused character of the amyloidosis signature (results listed in table 3, renamed table 2 in the revised version of the manuscript). Indeed, the lesions related to amyloidosis can be myocardial but also extra-myocardial, so that it is not interesting to focus on the myocardium for the analysis.
  • The authors should properly mention the axis titles in #Figure 3. For example, sensitivity (measuring unit).
  • (%) has been added in the axis titles
  • Please rewrite the Conclusions. It must be fully supported by the results reported and should include the major conclusions, the limitations of the work, and future work.
  • As asked by the referee 1, the conclusion has be modified, emphasizing the clinical significance and advantages of the work, scientific novelty and further research projects,

---------

Moreover, the assistant Editor, noticed that some sentences were similar with sentences we had already used in previous manuscripts.

These are fairly stereotypical methodological descriptions, identical to those of previous works.

These sentences, highlighted in yellow, have been reformulated (writing in green)

The two sections highlighted in yellow at the end of paragraph 4.2 and at the end of paragraph 4.4 do not appear in an earlier manuscript and therefore have not been modified.

Reviewer 2 Report

The manuscript entitled ‘Deep learning supplants visual analysis by experienced operators for the diagnosis of cardiac amyloidosis by cine-CMR’ utilizes a convolutional neural network (CNN) to discriminate cardiac amyloidosis from left ventricular hypertrophy (LVH) of other origin. Additionally, the authors have compared their findings with visual diagnosis assessed by experienced human operators. Overall, the manuscript is sound at conceptualization and writing part and the findings of the study are interesting. However, some minor concerns should be addressed by the authors before any possible consideration of this manuscript to be published in the journal ‘diagnostics.

  • In #Table 1 and table 2, if authors have taken the average value of some of the characteristics of sample population (for example, Weight and Height). It should be mentioned as ‘average weight’ and ‘average height’ in the tables.
  • Why the size of the images are not same in #Figure 1? The ' Image preparation ' section should mention the reason for cropping the image and taking the pictures of different magnifications.
  • The authors should properly mention the axis titles in #Figure 3. For example, sensitivity (measuring unit).
  • Please rewrite the Conclusions. It must be fully supported by the results reported and should include the major conclusions, the limitations of the work, and future work.

Author Response

(The authors gave the same response as above.)

Round 2

Reviewer 2 Report

Accept